# Relationship between Indel Variants within the *JAK2* Gene and Growth Traits in Goats

**DOI:** 10.3390/ani14131994

**Published:** 2024-07-06

**Authors:** Xian-Feng Wu, Qian Xu, Ao Wang, Ben-Zhi Wang, Xian-Yong Lan, Wen-Yang Li, Yuan Liu

**Affiliations:** 1Fujian Provincial Key Laboratory of Animal Genetics and Breeding/Institute of Animal Husbandry and Veterinary, Fujian Academy of Agricultural Sciences, Fuzhou 350013, China; wuxianfeng3080@163.com (X.-F.W.); xuqian2520@126.com (Q.X.); 2College of Animal Science, Fujian Agriculture and Forestry University, Fuzhou 350002, China; aodwywtd@163.com (A.W.); wbz5678@163.com (B.-Z.W.); 3Shaanxi Key Laboratory of Molecular Biology for Agriculture, College of Animal Science and Technology, Northwest A&F University, Yangling, Xianyang 712100, China; lanxianyong79@126.com

**Keywords:** goat, *JAK2* gene, insertion/deletion, growth traits, marker-assisted selection (MAS)

## Abstract

**Simple Summary:**

Janus kinase 2 (JAK2) plays a critical role in myoblast proliferation and fat deposition in animals, and our previous RNA-Seq analyses suggest that it is a candidate gene closely linked to muscle development. The effect of this gene on the body characteristics of animals has been investigated to a limited extent. Here, an analysis of 548 samples from three breeds examined the relationship between two insertion/deletion polymorphisms (del19008 and del72416) of the goat *JAK2* gene and body traits. The results showed that del19008 was associated with body height, body length, and trunk index. Locus del72416 was associated with BL, BH, TI, HuW, ChC, ChW, BLI, ChCI, CWI, and HuWI. In addition, the effect of the two InDels on the Nubian group was emphasized in terms of body height. Thus, these loci should be able to provide molecular markers for improving the body traits of goats.

**Abstract:**

*Janus kinase 2* (*JAK2*) plays a critical role in myoblast proliferation and fat deposition in animals. Our previous RNA-Seq analyses identified a close association between the *JAK2* gene and muscle development. To date, research delving into the relationship between the *JAK2* gene and growth traits has been sparse. In this study, we sought to investigate the relationship between novel mutations within the *JAK2* gene and goat growth traits. Herein, two novel InDel (Insertion/Deletion) polymorphisms within the *JAK2* gene were detected in 548 goats, and only two genotypes were designated as ID (Insertion/Deletion) and DD (Deletion/Deletion). The results indicate that the two InDels, the del19008 locus in intron 2 and del72416 InDel in intron 6, showed significant associations with growth traits (*p* < 0.05). Compared to Nubian and Jianzhou Daer goats, the del72416 locus displayed a more pronounced effect in the Fuqing breed group. In the Nubian breed (NB) group, both InDels showed a marked influence on body height (BH). There were strong linkages observed for these two InDels between the Fuqing (FQ) and Jianzhou (JZ) populations. The DD-ID diplotype was associated with inferior growth traits in chest width (ChW) and cannon circumference (CaC) in the FQ goats compared to the other diplotypes. In the NB population, the DD-DD diplotype exhibited a marked negative impact on BH and HuWI (hucklebone width index), in contrast to the other diplotypes. In summary, our findings suggest that the two InDel polymorphisms within the *JAK2* gene could serve as valuable molecular markers for enhancing goat growth traits in breeding programs.

## 1. Introduction

The Janus kinase/signal transducers and activators of transcription (JAK/STAT) pathway is an important intracellular signal transduction pathway [1]. Many JAK/STAT receptor agonists, such as growth hormone, prolactin, and gamma-interferon, send signals more directly to the nucleus than receptor tyrosine kinases [2,3]. Through in-depth research on the JAK2/STAT3 signaling pathway, it has been discovered that numerous cytokines (e.g., IFN, IL-2, IL-4, IL-6, and CNTF) and growth factors (e.g., EGF, PDGF, and CSF) utilize this pathway to induce cell proliferation, differentiation, or apoptosis [4]. Jak2 is typically activated by binding to type I or type II cytokine receptors and their corresponding ligands; it then phosphorylates STAT3, promoting its translocation to the nucleus and ultimately inducing the expression of target genes [5]. They play both specific and pleiotropic biological functions in embryonic development, immune function regulation, hematopoietic cell generation, and tumor occurrence [6]. This indicates the pivotal regulatory function of JAK2/STAT3 signaling in cellular processes [7].

Janus kinase (JAK) is a protein tyrosine kinase that plays an important role in the cytokine signaling pathway [8] and has four members, namely, JAK1, JAK2, JAK3, and TYK2. Jak2 plays a critical role in the signaling pathways of numerous cytokine receptors [9,10], which is a key transcription factor for myoblast proliferation and fat deposition [11,12]. A report showed that JAK2 is a GH receptor that undergoes ligand-dependent nuclear translocation and phosphorylates upon ligand binding with the GH receptor [13]. The disruption of hepatocyte growth GH signaling via the disruption of jak2 (JAK2L) leads to a fatty liver [14]. Following the administration of the jak inhibitor AG490 and transfection with JAK2-siRNA, there was a decline in the levels of *MHC*, *MGN*, *MEF2*, and *MyoD*. Additionally, the introduction of JAK2-siRNA markedly lowered the luciferase reporter gene activity that is dependent on *myogenin-*, *MRF-*, *MEF2-*, and *MyoD*. The expression of the *IGF2* gene and the *HGF* gene is also regulated by the *JAK2* gene [15].

Jak2 can influence the development of skeletal muscle and energy metabolism in organisms [16]. The *JAK2* gene was screened as a key candidate gene related to muscle growth and development in our previous RNA sequence [17]. It has been found that the *JAK2* gene is involved in the metabolism of body fat and that a decrease in the expression of this pathway leads to obesity [18]. Reports have revealed that mice with a liver-specific disruption of the GH signaling mediator JAK2 develop severe fatty livers and reduced body fat in mouse experiments [19]. JAK2 deficiency can lead to ischemia and the death of the embryo due to the failure of normal erythropoiesis in mice [20], and its abnormal activation also leads to the occurrence of blood diseases such as leukemia [21]. In cattle, JAK2 is related to lactation function, and it has been found that, when the expression of JAK2 rises, the activity of the JAK2-STAT signaling pathway increases, which can improve the lactation performance of dairy cows, such as milk protein production. Previous studies have demonstrated an association between the *JAK2* gene and growth traits in sheep [22] and cattle [23]. Hitherto, the *JAK2* gene has been considered a possible target for improving animal growth traits. As one of the many identified mutations, InDel is high in density and readily genotyped by fragment length polymorphisms. Genome-wide InDel polymorphisms among different accessions can be detected using whole-genome resequencing to guide the development of molecular markers. However, there have been no documented instances of InDel polymorphisms in the *JAK2* gene of goats. This study aimed to further detect the association between the novel InDel mutation within the goat *JAK2* gene and growth traits. In total, 548 samples from three breeds were used in this research, which may be useful for marker-assisted selection (MAS) in goat breeding.

## 2. Materials and Methods

The protocols (protocol number: 202207FJ002) of the Faculty of Animal Policy and Welfare Committee of the Fujian Academy of Agricultural Sciences (FAAS) for the use and care of animals in research were followed throughout all the experimental procedures.

### 2.1. Sample Collection and Zoometric Data Recording

Genomic DNA was extracted from the blood samples of all 548 healthy and body-mature female goats in the three breeds of Fuqing goat (FQ, *n* = 131), Nubian goat (NB, *n* = 296), and Jianzhou Daer goat (JZ, *n* = 121). The selected goats were approximately 2 years old and kept under the same diet and environmental conditions. All animals had their growth data collected, including body weight (BW), body height (BH), body length (BL), chest circumference (ChC), chest width (ChW), hucklebone width (HuW), chest depth (ChD), and cannon circumference (CaC). Gilbert et al. (1993) provided a method for measuring these qualities [24]. As recommended by the textbook written by Luo and Wang (1998) [25], the following metrics were also computed: trunk index (TI, Chest circumference/Body length × 100), body length index (BLI, Body length/Body height × 100), chest circumference index (ChCI, Chest circumference/Body height × 100), cannon circumference index (CaCI, Cannon circumference/Body height × 100), chest width index (CWI, Chest width/Chest depth × 100), and hucklebone width index (HuWI, Chest width/hucklebone width × 100).

### 2.2. DNA Separation and Creation of Genomic DNA Pools

Total DNA was isolated from the blood of 548 individuals using phenol–chloroform extraction, as described in Reference [24]. The DNA obtained was then diluted to a concentration of 50 ng/μL and stored at −80 degrees Celsius to facilitate the construction of genomic pools and enable polymorphism identification. In constructing the DNA pool, we randomly selected 50 samples for each breed, which were used to detect insertion-deletion mutations within the *JAK2* gene.

### 2.3. Primer Design, PCR Amplifications, and Genotyping

The sequence data and variation details of the *JAK2* gene (NC_030815.1) were obtained from the NCBI (Accessed on 7 July 2023. https://www.ncbi.nlm.nih.gov/) and Ensembl databases (Accessed on 8 July 2023. http://asia.ensembl.org/index.html). The primer pair for predicted mutation locus detecting was designed by the NCBI online primer design software (Accessed on 8 July 2023. https://www.ncbi.nlm.nih.gov/tools/primer-blast/index.cgi?LINK_LOC=Blast Home Primer) (Table 1). The DNA mixed pool was used to detect the novel InDel locus of the *JAK2* gene. The PCR products were sequenced by Sangon Biotec Co., Ltd. (Shanghai, China).

Each PCR reaction was performed in a 25 μL reaction mixture containing 50 ng genomic DNA; 0.5 µM of each primer; 1× buffer (including 1.5 mM MgCl_2_); 200 μM of the dNTPs (dATP, dTTP, dCTP, and dGTP); and 0.625 units of Taq DNA polymerase (TaKaRa, Dalian, China). The cycling protocol used in our previous study was followed [26]. Eventually, PCR products were detected via electrophoresis with 3.5% agarose gel stained with ethidium bromide at 120 V for 50 min. Two InDel variants were detected in the goat *JAK2* gene using primer pairs P4 and P13, namely, “del19008” and “del72416”, respectively.

### 2.4. Bioinformatics Analysis

The alignment of nucleotide and protein sequences was carried out with the assistance of MEGA software version 5.1 (Accessed on 20 July 2023. available at http://www.megasoftware.net/) and BioXM 2.6 (Nanjing Agricultural University, Nanjing, China). Evolutionary trees were generated through the neighbor-joining algorithm, as facilitated by MEGA version 5.1 and NCBI pairwise alignments (Accessed on 20 July 2023. http://www.ncbi.nlm.nih.gov/blast).

### 2.5. Statistical Analyses

We used the GDIcall online calculator (Accessed on 23 August 2023. http://www.msrcall.com/Gdicall.aspx) to calculate the genotypic and allelic frequencies of goat InDels in the *JAK2* gene and to examine the polymorphism information content (PIC). Hardy–Weinberg equilibrium (HWE) was tested using the *χ*^2^ test. Since only two genotypes were observed in each variant, an independent-sample *t*-test was used to analyze the correlation between the InDel and growth traits. The association between combined genotypes and growth traits was analyzed through ANOVA. SPSS statistical software (version 18.0) (IBM Corp., Armonk, NY, USA) was used for the calculations. *Y_ijk_
*=* μ *+* α_i_
*+* β_j_* + *H_k_
*+* ε_ijk_* was designed to investigate the association of the two InDels with growth traits, where *Y_ijk_* was the observation of the growth trait (body height, etc.) evaluated on the *i*th level of the fixed factor age (*α_i_*) and the *j*th level of the fixed factor genotype (*β_j_*); *μ* was the overall mean for each growth trait; *H_k_* was the fixed effect of the hatch; and *ε_ijk_* was the random error for the *ijk*th individual [26].

### 2.6. Linkage Disequilibrium (LD) and Combined Genotype Analysis

The linkage disequilibrium analysis of the loci del19008 and del72416 was conducted using Haploview 4.2 software (Accessed on 15 September 2023. http://www.broad.mit.edu/mpg/haploview/), in which D’ (0 < D’ < 1) and r^2^ (0 < r^2^ < 1) are indicators used to measure the strength of association across two linked genetic loci.

## 3. Results

### 3.1. Genotypic and Genetic Parameters of InDels within the Goat JAK2 Gene

Two novel deletion polymorphisms, del19008 and del72416, were detected in the *JAK2* gene using DNA pool sequencing, based on the predicted sequences from the European Variation Archive database (Accessed on 8 July 2023. http://www.ebi.ac.uk/eva/, Variant IDs: rs664724553 and rs670284366, respectively). Specifically, the del19008 locus is situated within the second intron of the *JAK2* gene on chromosome 8 of the goat (NC_030815.1: g.19008_19013delTCCAT), while the del72416 locus is found in the sixth intron of the gene (NC_030815.1: g.72416_72426delCTGTACGGTA). DNA gel electrophoresis revealed only two genotypes at these loci: DD (deletion/deletion) and ID (insertion/deletion), as shown in Figure 1.

### 3.2. Genetic Diversity of the Two InDels within the JAK2 Gene in Three Goat Populations

The genotypic and allelic frequencies, as well as polymorphism measures, were calculated based on genotype data (Table 2), including gene homozygosity (Ho), heterozygosity (He), effective allele numbers (Ne), and polymorphism information content (PIC). The chi-square (*χ*^2^) test was employed to assess whether the polymorphisms at loci del19008 and del72416 conformed to Hardy–Weinberg equilibrium (HWE). The results revealed that, in the three goat populations, the frequency of the ID genotype was higher than that of the DD genotype for both loci. Accordingly, the frequency of the “D” allele was greater than that of the “I” allele. The genetic diversity at both loci was moderate (0.25 < PIC < 0.5) in the tested populations, as indicated by their PIC values (Table 2). The *χ*^2^ test results indicated that neither of the two mutation loci was in HWE in the three populations (*p* < 0.05).

### 3.3. Bioinformatics Analysis of JAK2

The goat JAK2 mRNA sequence was found to encode a protein consisting of 1132 amino acids. We observed that the coding sequence of goat JAK2 shares 99%, 99%, 99%, 99%, 99%, 96%, 93%, 94%, 96%, and 94% similarity with *Ovis aries* (XP_042099355.1, 1132 amino acids), *Bos taurus* (XP_005210038.1, 1132 amino acids), *Bos indicus* (XP_019821553.1, 1132 amino acids), *Bos mutus* (ELR59526.1, 1132 amino acids), *Bubalus bubalis* (XP_006068266.1, 1132 amino acids), *Camelus dromedarius* (XP_031305623.1, 1130 amino acids), *Mus musculus* (NP_032439.2, 1132 amino acids), *Rattus rattus* (XP_032747668.1, 1132 amino acids), *Sus scrofa* (XP_020938465.1, 1131 amino acids), and *Homo sapiens* (NP_001309125.1, 1132 amino acids). The phylogenetic analysis conducted using the goat JAK2 sequence revealed a close evolutionary relationship with sheep JAK2 while indicating the least similarity with the clade embodying sequences from *R. rattus* and *M. musculus* (Figure 2).

### 3.4. Association between InDel Variants and Growth Traits in Goats

The analysis of the association between the InDel locus and phenotypic growth characteristics revealed a significant correlation with several growth traits in goats (*p* < 0.05) (Table 3). For the del19008 locus, individuals with ID genotypes displayed a more superior growth trait than DD with BH (*p* = 0.014) and TI (*p* = 0.009) in the group of NB and with HuW (*p* = 0.013) in the FQ breed, as well as CWI (*p* = 0.019) in the JZ population. The del72416 locus was highly associated with growth traits, including BL (*p* = 0.019), HuW (*p* = 0.011), ChC (*p* = 0.016), and ChW (*p* = 0.002) in the FQ goats and BH (*p* = 0.004), BLI (*p* = 0.00004), ChCI (*p* = 0.017), CWI (*p* = 0.046), and HuWI (*p* = 0.004) of the NB breed. In summary, the ID genotype was associated with more favorable growth traits at the del19008 locus, whereas, at the del72416 locus, the DD genotype was associated with better growth traits than the ID genotype.

### 3.5. LD and Haplotype Analysis

In the FQ and JZ populations, a strong LD was found between the loci del19008 and del72416, as indicated by the correlation coefficients (FQ: D’ = 1.000, r^2^ = 0.518; JZ: D’ = 1.000, r^2^ = 0.326), while moderate LD was observed in the NB population (D’ = 0.333, r^2^ = 0.108) (Figure 3). Subsequently, a haplotype association analysis was conducted with a minimum frequency criterion of 0.001 for haplotype consideration (Table 4). Among the four identified haplotypes, the ID-ID genotype exhibited the highest frequency in both the FQ and NB populations. Correlation analyses of the four combined genotypes in the FQ and NB goats indicated that the DD-ID diplotype was associated with significantly inferior growth traits than the other haplotypes in ChW (*p* = 0.003) and CaC (*p* = 0.011) in FQ goats. In the NB population, the DD-DD diplotype showed extremely significant inferiority in BH (*p* = 0.001) and HuWI (*p* = 6 × 10^−7^) compared to the other diplotypes. Conversely, the DD-DD diplotype was associated with extremely significant superiority in the BLI (*p* = 1 × 10^−6^) and ChCI (*p* = 6 × 10^−7^) compared to the other diplotypes. These findings provide further evidence for the significant role of the *JAK2* gene in influencing growth traits in both FQ and NB goat breeds.

## 4. Discussion

Although identifying candidate genes holds the potential to improve economically important traits in livestock, the primary genes or quantitative trait loci (QTL) involved in these traits are still not well defined [27,28]. Consequently, identifying key genes and their genetic variations is essential to lay the theoretical groundwork for applying molecular-marker-assisted selection (MAS) in goat breeding.

Previous research has demonstrated that InDel markers within the *ATBF1*, *STAT5A*, *CPT1a*, and *CFAP43* genes significantly influence growth traits in goats [25,29,30,31]. Our previous RNA sequencing results indicated a relationship between the *JAK2* gene and growth traits in FQ and NB breeds. In this study, two InDel loci within the *JAK2* gene, del19008 and del72416, were identified across an indigenous breed (FQ) and two introduced breeds (NB and JZ). These sequences deviate from those listed in the European Variation Archive (EVA) database (Accessed on 7 July 2023. http://www.ebi.ac.uk/eva/), suggesting breed-specific variances. The sequences in the EVA are primarily based on data from African dwarf goats, which are subject to unique environmental conditions.

Jak2 is critically involved in myoblast proliferation and fat deposition in animals [11,32]. In adipose tissue, Jak2 exhibits epistasis over the liver regarding insulin sensitivity and response. The inhibition of hepatocyte GH signaling at the level of the GH receptor, Jak2, or stat5 elevates the circulating GH levels, which could exacerbate GH-mediated processes such as lipolysis in adipose tissue [14]. Moreover, an expanding body of research is exploring the molecular mechanisms of jak2 function in muscle tissues [33,34]. In particular, studies have found that the JAK2/STAT2/STAT3 pathway regulates the expression of *HGF* and *IGF2*, which are crucial for myoblast proliferation and differentiation [35]. Jak2 is essential for development, and its conservation across mammals has been established [35,36]. Conventional knockout mouse models show postnatal or embryonic lethality [8], possibly explaining the absence of homozygous deletions for the two InDels in the three goat populations. A protein similarity analysis indicated that goat JAK2 shares over 94% similarity with ten other species, highlighting its high conservation in ruminants. Screening results for InDel loci revealed only two out of seventeen predicted loci (Table 1), further indicating the conservative nature of the *JAK2* gene. These findings underscore the vital role of *JAK2* in the developmental processes of goats.

The genotype frequencies of the two InDels were found to be similar among the FQ, NB, and JZ goats (Table 2). The two InDels exhibited two genotypes: homozygous wild-type (DD) and heterozygous genotype (ID). The “D” allele was the predominant allele in all three goat breeds. The absence of the homozygous mutant genotype (II) was evident in the results of the HWE test. Moreover, a significant departure from HWE was noted in these breeds, likely due to non-random mating resulting from population stratification, selection, or genetic drift [37]. The two InDels showed strong linkage in the FQ and JZ populations and moderate linkage disequilibrium in the NB group. The Fuqing goat is a distinct local breed from Fujian Province, China [38], while the Nubian goat originates from Egypt. The Jianzhou Daer goat is a cross between local Jianzhou and Nubian goats [39]. It can be hypothesized that recombination between del19008 and del72416 occurred during the evolutionary selection process in these breeds.

In molecular breeding, the pivotal economic traits of growth and body weight performance are of increasing importance. Jak2 is an integral part of the growth hormone receptor pathway, involved in transmitting signals from class II cytokine receptors [5], and plays a critical role in the growth, development, and reproductive traits of ruminants [23]. Previous research has mostly concentrated on the impact of SNPs within the *JAK2* gene on cattle and sheep traits, such as growth [40], milk production [41], and mastitis resistance [42]. However, the association between InDel polymorphisms of the *JAK2* gene and growth traits in goats has been less investigated. This study aimed to address this gap by examining the association of two InDel polymorphisms of the *JAK2* gene with growth traits in goats. The del19008 polymorphism was found to primarily affect the BH, HuW, and other traits in both FQ and NB breeds (Table 4), whereas the del72416 polymorphism mainly impacted the BH, BL, HuW, ChC, ChW, and other traits in the same breeds. Nevertheless, these InDels did not have a significant effect on the JZ group. The del72416 locus’s influence was more pronounced in the FQ group compared to the NB and JZ groups, particularly with traits like the ChW, BL, HuW, and ChC exhibiting higher values in the DD genotype. Given that the FQ breed is more primitive and the NB and JZ breeds are more selectively bred, such differences might be attributed to breeding pressures. Notably, in the NB group, the InDels’ effects on body height were particularly pronounced. In both InDels, the ID phenotypic values significantly surpassed those of DD, which could be due to heterosis. In summary, we hypothesize that the DD genotypes at the del72416 locus might confer higher feed conversion efficiency during Fuqing goat development. InDel variants can directly influence gene expression and function, including impacts on mRNA translation when located in exons [43] and transcription factor binding when occurring in introns [44,45,46]. The presence of enhancer elements within introns and their potential regulatory impact on gene expression should particularly be taken into account [47,48]. Given that a substantial number of InDels in gene intronic regions can significantly affect animal production traits, such as the notable 11 bp deletion within intron 9 of the *CSN1S1* gene shown to influence milk yield and body measurements in goats [49,50], the newly identified InDels of the *JAK2* gene could serve as valuable genetic markers for growth trait selection in goat breeding.

## 5. Conclusions

The del19008 and del72416 loci of the *JAK2* gene were explored, and only two genotypes were designated in three goat breeds. The analyses revealed significant associations between these two loci and growth traits. The influence of the del72416 locus on growth traits was notably stronger in the FQ breed compared to the NB and JZ breeds. Additionally, in the NB population, the deleterious effects of the two InDels were primarily manifested in the BH. Moreover, the DD-ID diplotype was associated with inferior growth traits in the ChW and CaC in the FQ goats compared to the other diplotypes. In the NB population, the DD-DD diplotype exhibited a marked negative impact on the BH and HuWI, in contrast to the other diplotypes. Conversely, the DD-DD diplotype was associated with extremely significant superiority in the BLI and ChCI compared to the other diplotypes. These findings suggest that the two InDel polymorphisms significantly influence growth traits and could serve as valuable DNA markers for MAS in breeding programs aimed at enhancing growth characteristics in goats.

## Figures and Tables

**Figure 1 animals-14-01994-f001:**
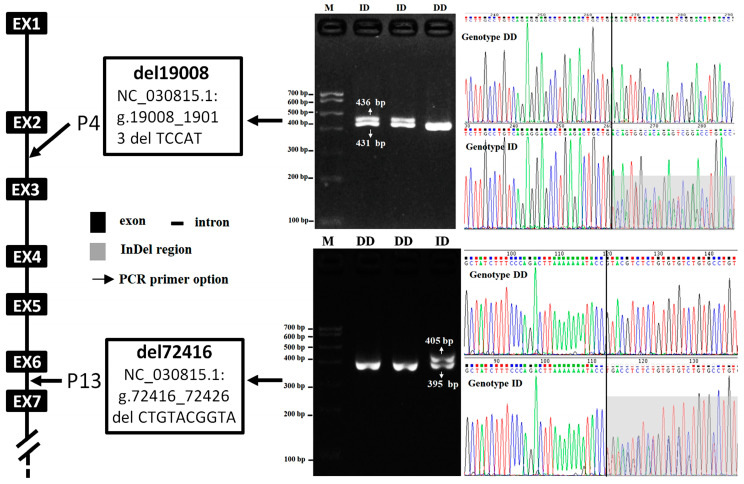
Electrophoresis diagrams and sequencing diagrams of the loci del19008 and Del72416 in the goat *JAK2* gene. Note: M = molecular marker; del19008 genotypes (DD = 431 bp, ID = 431 bp and 436 bp): the area with the gray box is the sequence of the 5 bp deletion; del72416 genotypes (DD = 395 bp, ID = 395 bp and 405 bp): the region with the gray box is the bimodal sequence of the 10 bp deletion.

**Figure 2 animals-14-01994-f002:**
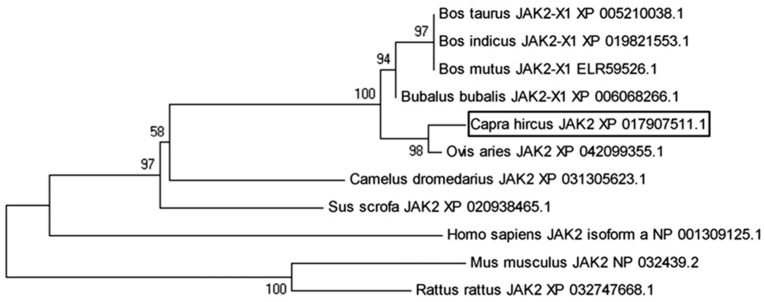
Phylogenetic analysis of JAK2 in 11 species. Note: The tree was constructed from the amino acid sequences via the neighbor-joining method using MEGA 5.1. The numbers on the joints are values from the bootstrap test, and the branch length represents the evolutionary time.

**Figure 3 animals-14-01994-f003:**
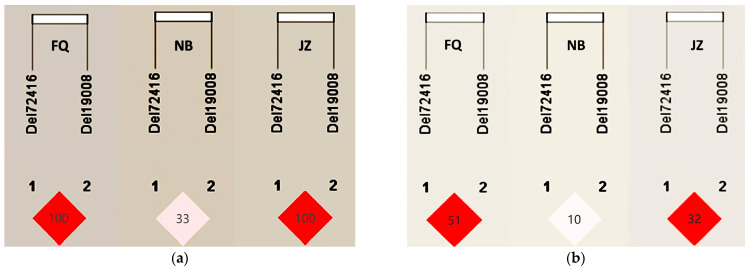
Linkage disequilibrium plot of two InDel loci of the *JAK2* gene in goats. (**a**) D’ value of three goat breeds; (**b**) r^2^ value of three goat breeds.

**Table 1 animals-14-01994-t001:** Primer information.

Loci	Primer Sequences (5′→3′) (Nucleotide Position)	Sizes (bp)	Rs Number	Polymorphism
P1	F: ACAGGCAGGGCTCATTGTTG	426	rs670502347	No polymorphism
	R: ACTCAGCTGCCCCTATCCTT			
P2	F: CTGGGCTTGTAGTTTGGATTTGT	639	rs656761347	No polymorphism
	R: GCAGTCGTTAATTATGCATGGGA			
P3	F: AAGATCATTATGGCCTGGAT	414	rs652471794	No polymorphism
	R: CTGGCATTTGAGCATTTGTT			
P4	F: AGTTTTGAAGACATGGTTGT	431/436	Del19008 rs664724553	**Polymorphism**
	R: ACCTCCAGACAGTATTGCTA			
P5	F: TCCTATCATCGTCTATCACCA	452	rs646322931	No polymorphism
	R: ATATGATCCAGCAGTTCCACT			
P6	F: TTATGTATCTTTAGGTTTCC	441	rs688366678	No polymorphism
	R: ACAAGAAATACTAAAAGGAG			
P7	F: ACACTGGCAATGTGACTTATTT	415	rs658809227	No polymorphism
	R: GTCTTCGTTACCAGGATGAG			
P8	F: AGTTGTGCTAGTTTACGTTTCT	444	rs658982284	No polymorphism
	R: GGTTACAGTTTATGGGGTCA			
P9	F: AGAAAATGAGGAAACCAAAAGTT	470	rs649684777	No polymorphism
	R: GGAAATAGAGGAGGGCAGAG			
P10	F: TTTGCGAAACTTTTCCTAGT	327	rs660232567	No polymorphism
	R: CAGCCTATTTCACTGATTCTAC			
P11	F: CACTGCCAGTCCATCCATTTT	399	rs656433773	No polymorphism
	R: ATGTCACTTTGCACCCACCA			
P12	F: TTCTGGTGGGTGCAAAGT	459	rs652577437	No polymorphism
	R: AGACAACCTACTGAATGGGATA			
P13	F: CAGGATTGATGGAAGAACCG	395/405	Del72416 rs670284366	**Polymorphism**
	R: GAGCAGATGAAGAATGAAACAGAA			
P14	F: GATTGAATAATGGGGTTGGT	402	rs660426344	No polymorphism
	R: CATTTCTTTGGCTACTTTGC			
P15	F: TACCATGTGCAGGCAACTAT	336	rs684000928	No polymorphism
	R: TAGGGATTACCTGGGATTGG			
P16	F: TCTCGTCAAGGCTATGGTTTT	461	rs671844323	No polymorphism
	R: CTGGGTTCAGTTCAGTTCAGTT			
P17	F: CATTGTATTATACTTTAGCCTTATT	587	rs661217484	No polymorphism
	R: AATCTGATTACTGGCTGAC			

Note: NCBI reference sequence NC_030815.1.

**Table 2 animals-14-01994-t002:** Genotypic and allelic frequencies and genetic parameters of InDels within the *JAK2* gene.

Locus	Breeds (Size)	Genotypic Frequencies (Individual Quantity)		Allelic Frequencies		Ho	He	PIC	HWE
		DD	ID	D	I				*p*-Value
Del19008	FQ (*n* = 129)	0.186 (24)	0.814 (105)	0.593	0.407	0.517	0.483	0.366	*p* < 0.05
	NB (*n* = 288)	0.490 (141)	0.510 (147)	0.745	0.255	0.620	0.380	0.308	*p* < 0.05
	JZ (*n* = 120)	0.333 (40)	0.667 (80)	0.667	0.333	0.556	0.444	0.346	*p* < 0.05
Del72416	FQ (*n* = 125)	0.128 (16)	0.872 (109)	0.564	0.436	0.508	0.492	0.371	*p* < 0.05
	NB (*n* = 288)	0.479 (138)	0.521 (150)	0.740	0.260	0.615	0.385	0.311	*p* < 0.05
	JZ (*n* = 120)	0.208 (25)	0.792 (95)	0.604	0.396	0.522	0.478	0.364	*p* < 0.05

**Table 3 animals-14-01994-t003:** Relationship between the InDels of the *JAK2* gene and growth traits in goats.

Locus	Breeds	Growth Traits	Mean ± SE		*p*-Value
			DD	ID	
del19008	FQ	HuW (cm)	14.36 ± 0.34 ^b^	15.17 ± 0.13 ^a^	0.013
	NB	BH (cm)	68.05 ± 0.48 ^b^	69.70 ± 0.46 ^a^	0.014
		TI	131.51 ± 0.76 ^B^	134.46 ± 0.83 ^A^	0.009
		BLI	96.88 ± 0.61 ^A^	93.03 ± 0.61 ^B^	0.00001
		ChCI	127.09 ± 0.78 ^a^	124.73 ± 0.81 ^b^	0.036
	JZ	CWI	57.3 ± 0.74 ^b^	59.74 ± 0.70 ^a^	0.019
del72416	FQ	BL (cm)	34.80 ± 1.65 ^a^	30.43 ± 0.66 ^b^	0.019
		HuW (cm)	57.66 ± 1.37 ^a^	54.60 ± 0.40 ^b^	0.011
		ChC (cm)	77.74 ± 1.53 ^a^	73.39 ± 0.64 ^b^	0.016
		ChW (cm)	17.61 ± 0.46 ^A^	16.48 ± 0.17 ^B^	0.002
	NB	BH (cm)	67.90 ± 0.45 ^B^	69.8 ± 0.49 ^A^	0.004
		BLI	96.80 ± 0.62 ^A^	93.19 ± 0.61 ^B^	0.00004
		ChCI	127.93 ± 0.77 ^A^	124.00 ± 0.79 ^B^	0.0004
		CWI	59.82 ± 0.52 ^b^	61.49 ± 0.65 ^a^	0.046
		HuWI	110.32 ± 1.14 ^B^	115.89 ± 1.51 ^A^	0.004

Note: Values bearing distinct superscripts in an identical column display significant differences, where *p* < 0.05 is denoted by (a,b) and *p* < 0.01 by (A,B). Body height (BH), body length (BL), chest circumference (ChC), chest width (ChW), hucklebone width (HuW), body length index (BLI), chest circumference index (ChCI), chest width index (CWI), and hucklebone width index (HuWI).

**Table 4 animals-14-01994-t004:** Combined effect of two loci on growth traits (means ± standard errors) in goats.

Breed	Growth Traits	Combined Genotypes (Mean ± SE)				*p*-Values
		DD-DD	DD-ID	ID-DD	ID-ID	
FQ	ChW (cm)	15.95 ± 0.92 ^AB^ (*n* = 4)	15.71 ± 0.55 ^B^ (*n* = 20)	18.16 ± 0.45 ^Aa^ (*n* = 12)	16.60 ± 0.17 ^ABb^ (*n* = 87)	0.003
	CaC (cm)	14.80 ± 0.55 ^AB^ (*n* = 4)	14.27 ± 0.40 ^Bc^ (*n* = 20)	15.99 ± 0.37 ^Aa^ (*n* = 12)	15.09 ± 0.15 ^ABb^ (*n* = 87)	0.011
NB	BH (cm)	66.84 ± 0.51 ^B^ (*n* = 80)	70.35 ± 0.85 ^A^ (*n* = 60)	69.55 ± 0.74 ^A^ (*n* = 57)	69.46 ± 0.59 ^A^ (*n* = 90)	0.001
	BLI	98.71 ± 0.73 ^A^ (*n* = 80)	93.12 ± 0.80 ^B^ (*n* = 60)	93.65 ± 0.91 ^B^ (*n* = 57)	93.44 ± 0.90 ^B^ (*n* = 90)	0.000001
	ChCI	130.87 ± 1.02 ^A^ (*n* = 80)	123.48 ± 1.10 ^B^ (*n* = 60)	124.05 ± 1.11 ^B^ (*n* = 57)	124.24 ± 1.03 ^B^ (*n* = 90)	0.0000006
	HuWI	108.03 ± 1.59 ^B^ (*n* = 80)	120.96 ± 2.50 ^Aa^ (*n* = 60)	111.54 ± 1.75 ^Ab^ (*n* = 57)	113.82 ± 1.77 ^Aab^ (*n* = 90)	0.0000006

Note: Values with different superscripts within the same column differ significantly at *p* < 0.05 (a,b,c) and *p* < 0.01 (A,B); the same letters or no letters indicate no significant difference. Body height (BH), chest width (ChW), cannon circumference (CaC), body length index (BLI), chest circumference index (ChCI), and hucklebone width index (HuWI).

## Data Availability

The data presented in this study are available on request from the corresponding author.

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
