# Peer review of "Relationship between Indel Variants within the JAK2 Gene and Growth Traits in Goats"

_animals, 2024, doi:10.3390/ani14131994_

Round 1

Reviewer 1 Report

Comments and Suggestions for Authors

Comments: The manuscript entitled “Relationship between indel variants within the JAK2 gene and growth traits in goat investigates the association between two indels of the JAK2 gene and growth traits in goats. Overall, the manuscript is poorly organized, some of the content is not easy to understand, and there are some mismatches. The major concerns are as follows:

1.     In the introduction, most of the content describes the progress of research on the JAK2 gene. I think it is appropriate to add some current deficiencies, as well as the purpose and significance of this study.

2.     Lines 85-88, please reorganize the language.

3.     In the Materials and Methods, are the animals used male or female? How old were they? How old were they at the time of measurement? Were these animals kept under the same conditions? Please add this information.

4.     In 2.2. DNA separation and creation of genomic DNA pools. What are the quality requirements for DNA used for sequencing?

5.     In Statistical analyses. Are these animals all from the same batch, and if not, I think the batch effect should also be taken into account in the general linear model.

6.     Specific information on sequencing and genotyping should be added.

7.     Line 156, what does the blue font stand for?

8.     Line 179, please standardize your writing.

9.     Line 186, Figure 2?

10.  Line 195, Table 4? Change “…Individuals” to “individuals”

11.  Lines 195-196, “Individuals with ID genotypes were larger than those with ID with BH”. Please check carefully.

12.  I suggest verifying the differences in the expression of the candidate genes in individuals of different genotypes by qPCR to enhance the conclusions.

13.  Line 239-240, please reorganize the language.

14.  Note that genes should be italicized.

15.  Besides the main points above, I would suggest that the manuscript needs to be corrected by a native English speaker.

Comments on the Quality of English Language

I would suggest that the manuscript needs to be corrected by a native English speaker.

Author Response

Comments 1:

The manuscript entitled “Relationship between indel variants within the JAK2 gene and growth traits in goat” investigates the association between two indels of the JAK2 gene and growth traits in goats. Overall, the manuscript is poorly organized, some of the content is not easy to understand, and there are some mismatches. The major concerns are as follows:

  1. In the introduction, most of the content describes the progress of research on the JAK2 gene. I think it is appropriate to add some current deficiencies, as well as the purpose and significance of this study.

Response: Thanks for your good suggestion. We have revised this sentence, and added the current deficiencies about the research of the Indel polymorphisms in the JAK2 gene.

Line 85-93. “Hitherto, the JAK2 gene has been considered a possible target for improving animal growth traits. As one of the many identified mutations, InDel is high in density and readily genotyped by fragment length polymorphisms. Genome-wide InDel poly-morphisms among different accessions can be detected using whole-genome resequencing to guide the development of molecular markers. However, there have been no documented instances of InDel polymorphisms in the JAK2 gene of goats. This study aimed to further detect the association between the novel InDel mutation within the goat JAK2 gene and growth traits. In total, 548 samples from three breeds were used in this research, which may be useful for marker-assisted selection (MAS) in goat breeding.”

  1. Lines 85-88, please reorganize the language.

Response: Thanks for your good suggestion. We have revised the sentences.

Line 89-93. “However, there have been no documented instances of InDel polymorphisms in the JAK2 gene of goats. This study aimed to further detect the association between the novel InDel mutation within the goat JAK2 gene and growth traits. In total, 548 samples from three breeds were used in this research, which may be useful for marker-assisted selection (MAS) in goat breeding.”

  1. In the Materials and Methods, are the animals used male or female? How old were they? How old were they at the time of measurement? Were these animals kept under the same conditions? Please add this information.

Response: I’m sorry for my mistake. It has revised in the part of Materials and Methods.

Line 100-103. “Genomic DNA was extracted from the blood samples of all 548 healthy and body-mature female goats in three breeds of Fuqing goat (FQ, n=131), Nubian goat (NB, n=296), and Jianzhou Daer goat (JZ, n=121). The selected goats were approximately 2 years old and had been kept under the same diet and environmental conditions.”

  1. In 2.2. DNA separation and creation of genomic DNA pools. What are the quality requirements for DNA used for sequencing?

Response: The concentration and quality of isolated genomic DNA was quantified using a NanoDrop 1000 spectrophotometer (Thermo Fisher Scientific Inc., Wilmington, DE, USA). The values of 260/280 and 260/230 should be range 1.8 to 2.0 and only one band in 1% agarose electrophoresis.

  1. In Statistical analyses. Are these animals all from the same batch, and if not, I think the batch effect should also be taken into account in the general linear model.

Response: Thanks for your good suggestion. The collected animals are not from the same batch. We have revised the general linear model and added the batch effect.

Line 151-158. “The association between combined genotypes and growth traits was analyzed through an ANOVA. SPSS statistical software (version 18.0) (IBM Corp., Armonk, NY, USA) was used for calculations. Yijk =μ + αi + βj +Hk+ εijk was designed to investigate the association of the two InDels with growth traits, where Yijk was the observation of the growth trait (body height, etc.) evaluated on the ith level of the fixed factor age (αi) and the jth level of the fixed factor genotype (βj); μ was the overall mean for each growth trait; Hk was the fixed effect of hatch; and εijk was the random error for the ijkth individual [26].”

  1. Specific information on sequencing and genotyping should be added.

Response: Thanks for your good suggestion. The information was added in Line 130-136. “2.3. Primer design, PCR amplifications, and genotyping”.

  1. Line 156, what does the blue font stand for?
    Response: I’m sorry for my mistake. It has revised in the manuscript.

  1. Line 179, please standardize your writing.

Response: Thanks for your good suggestion. We have revised the sentences.

Line 194-205. “The goat JAK2 mRNA sequence was found to encode a protein consisting of 1132 amino acids. We observed that the coding sequence of goat JAK2 shares 99%, 99%, 99%, 99%, 99%, 96%, 93%, 94%, 96%, and 94% similarity with Ovis aries (XP_042099355.1, 1132 amino acids), Bos taurus (XP_005210038.1, 1132 amino acids), Bos indicus (XP_019821553.1, 1132 amino acids), Bos mutus (ELR59526.1, 1132 amino acids), Bubalus bubalis (XP_006068266.1, 1132 amino acids), Camelus dromedarius (XP_031305623.1, 1130 amino acids), Mus musculus (NP_032439.2, 1132 amino acids), Rattus rattus (XP_032747668.1, 1132 amino acids), Sus scrofa (XP_020938465.1, 1131 amino acids), and Homo sapiens (NP_001309125.1, 1132 amino acids). Phylogenetic analysis conducted using the goat JAK2 sequence revealed a close evolutionary relationship with sheep JAK2, while indicating the least similarity with the clade embodying sequences from R. rattus and M. musculus (Figure 2).”

  1. Line 186, Figure 2?

Response: I’m sorry for my mistake. It has revised in the manuscript. The Figure 2 has change to Phylogenetic analysis.

  1. Line 195, Table 4? Change “…Individuals” to “individuals”

Response: I’m sorry for my mistake. It has revised in the manuscript.

  1. Lines 195-196, “Individuals with ID genotypes were larger than those with ID with BH”. Please check carefully.

Response: I’m sorry for my mistake. It has revised in the manuscript.

Line 213-216. “For the del19008 locus, individuals with ID genotypes displayed a more superior growth trait than DD with BH (P = 0.014) and TI (P = 0.009) in the group of NB, and with HuW (P = 0.013) in the FQ breed, as well as CWI (P = 0.019) in the JZ population.”

  1. I suggest verifying the differences in the expression of the candidate genes in individuals of different genotypes by qPCR to enhance the conclusions.

Response: Thank the reviewers for these precious comments and suggestions. We acknowledge that incorporating qPCR expression analysis might enhance the robustness of our conclusions. Regrettably, there are not enough sample materials available for such qPCR experiments at this time. Additionally, we face a selection dilemma as to which type of tissue would be most appropriate for qPCR, since an InDel locus's effect on gene expression may not be consistently mirrored across different tissues. This variability complicates any direct correlation between the InDel locus and growth traits within a single tissue type analysis. As a result, we regretfully inform you that we are unable to address your suggestion currently. However, we recognize the importance of this method and will condor it exploring it in future research endeavors.

Once again, we thank the reviewers for their valuable comments and guidance.

  1. Line 239-240, please reorganize the language.

Response: I’m sorry for my mistake. I have rewritten the sentence.

Line 255-256. “Previous research has demonstrated that InDel markers within the ATBF1, STAT5A, CPT1a, and CFAP43 genes significantly influence growth traits in goats [25,29-31].”

  1. Note that genes should be italicized.

Response: I’m sorry for my mistake. It has revised in the manuscript.

  1. Besides the main points above, I would suggest that the manuscript needs to be corrected by a native English speaker.

Response: Thanks for your good suggestion. This manuscript had been revised by the English editing service of MDPI.

Reviewer 2 Report

Comments and Suggestions for Authors

The presented studies are relatively innovative. Results might be useful for MAS selection in goats with desirable growth qualities. However, there are similar studies investigated InDels of the JAK/STAT genes in sheep.

The manuscript is well done and complete in its experimental design.

Some minor points:

Editorial remarks: lines 33, 34, 46, 47 please add explanation of abbreviations; line 44 add the abbreviation for growth hormone; lines 60, 62 remove the abbreviation GH; line 238 remove the abbreviations MAS.

Please standardize gene notation throughout the text.

In the Materials and Methods section there is no information about the gender and age of the goats. Please fill in the missing data.

Author Response

Comments 2:

The presented studies are relatively innovative. Results might be useful for MAS selection in goats with desirable growth qualities. However, there are similar studies investigated InDels of the JAK/STAT genes in sheep.

The manuscript is well done and complete in its experimental design.

Some minor points:

Editorial remarks: lines 33, 34, 46, 47 please add explanation of abbreviations; line 44 add the abbreviation for growth hormone; lines 60, 62 remove the abbreviation GH; line 238 remove the abbreviations MAS.

Response: Thanks for your good suggestion. It has revised in the manuscript.

In abstract. “two novel InDel (Insertion/Deletion) polymorphisms within the JAK2 gene were detected in 548 goats, and only two genotypes were designated as ID (Insertion/Deletion) and DD (Deletion/Deletion).”

“In the Nubian breed (NB) group, both InDels showed a marked influence on body height (BH). There were strong linkages observed for these two InDels between the Fuqing (FQ) and Jianzhou (JZ) populations. The DD-ID diplotype was associated with inferior growth traits in chest width (ChW) and cannon circumference (CaC) in the FQ goats compared to the other diplotypes.”

Line 48-49. “Many JAK/STAT receptor agonists, such as growth hormone, prolactin, and gamma-interferon, send signals more directly to the nucleus than do receptor tyrosine kinases”

Please standardize gene notation throughout the text.

Response: I’m sorry for my mistake. It has revised in the manuscript.

In the Materials and Methods section there is no information about the gender and age of the goats. Please fill in the missing data.

Response: I’m sorry for my mistake. It has revised in the part of Materials and Methods.

Line 100-103. “Genomic DNA was extracted from the blood samples of all 548 healthy and body-mature female goats in three breeds of Fuqing goat (FQ, n=131), Nubian goat (NB, n=296), and Jianzhou Daer goat (JZ, n=121). The selected goats were approximately 2 years old and had been kept under the same diet and environmental conditions.”

Reviewer 3 Report

Comments and Suggestions for Authors

The MS entitled Relationship between indel  written by Wu et al analyzed the polymorphisms of JAK2 gene, the genetic characteristics in three goat populations, the relationship with the growth traits in goats. Results showed that 2 InDels in the intron section were found and the variants affect the growth traits including body height, body length, hucklebone width, chest circumference, chest width and their relative indexes in three breeds of goats to an extent. The results will provide a basis for the scientific breeding of goats and lay a foundation for the further study of the mechanism of JAK2 in goat growth.

General Comments

The studies reported in this manuscript were conducted in an appropriate fashion with objectives being addressed using appropriate study designs and methods to achieve the objectives. The written quality of the manuscript, however, can be markedly improved and the authors should endeavor to do so if this manuscript is to be further considered for publication in Animals.

The most important is the results seems not completely consistent with that in the abstract, the same as the content in the text with that in the table.

The age of the body traits detected is not clear. Three breeds were used in the current study, what farms did the goats from? How about the feeding conditions or feeding standard? When did the data for body size were collected?

Specific Comments

L25 abstract: pls tell the information about the genetic characteristics of the two loci, and the Haplotype and its combination on the growth traits

L33: both loci have significant associations for all parameters? Seems not

L35 compared with

L37 body height

L38 the incomplete conclusion, the sentence is not completed

L63 jak2 change to JAK2

L86 repeated goat

L102 pls give a brief introduction of the relationship between the body size and its index from the different references?

L131and L143: pls explain the Statistical analysis of the combined genotypes with the growth traits

L191: Table 3. The accession numbers and details of the amino acid sequences employed in constructing the tree, this table is not necessary and can be deleted. The authors can describe the length of the AA sequences in the text.

L195: Individuals with ID genotypes were larger than those with ID with ...? what does the author want to tell the readers? not complete or clear descriptions, which is also not consistent with those in the table 4.

L228: Table 5, regarding to the letters indicating the differences, they are not marked correctly. eg. the first line (FQ, ChW), should be AB, B, Aa, ABb? Pls check them especially for the FQ breed, BH and HuWI in NB breed, and explain it in the legends: the same letters or no letters indicate no significant difference.

Comments on the Quality of English Language

 Moderate editing of English language required

Author Response

Comments 3:

The MS entitled “Relationship between indel …” written by Wu et al analyzed the polymorphisms of JAK2 gene, the genetic characteristics in three goat populations, the relationship with the growth traits in goats. Results showed that 2 InDels in the intron section were found and the variants affect the growth traits including body height, body length, hucklebone width, chest circumference, chest width and their relative indexes in three breeds of goats to an extent. The results will provide a basis for the scientific breeding of goats and lay a foundation for the further study of the mechanism of JAK2 in goat growth.

General Comments

The studies reported in this manuscript were conducted in an appropriate fashion with objectives being addressed using appropriate study designs and methods to achieve the objectives. The written quality of the manuscript, however, can be markedly improved and the authors should endeavor to do so if this manuscript is to be further considered for publication in Animals.

The most important is the results seems not completely consistent with that in the abstract, the same as the content in the text with that in the table.

Response: I’m sorry for my mistake. It has revised in the manuscript; I have rewritten the abstract and carefully check the “3.4. Association between InDel variants and growth traits in goat”, “3.5. LD and haplotype analysis” and Table 3 and 4.

The age of the body traits detected is not clear. Three breeds were used in the current study, what farms did the goats from? How about the feeding conditions or feeding standard? When did the data for body size were collected?

Response: I’m sorry for my mistake. It has revised in the part of Materials and Methods.

Line 100-103. “Genomic DNA was extracted from the blood samples of all 548 healthy and body-mature female goats in three breeds of Fuqing goat (FQ, n=131), Nubian goat (NB, n=296), and Jianzhou Daer goat (JZ, n=121). The selected goats were approximately 2 years old and had been kept under the same diet and environmental conditions.”

Specific Comments

  1. L25 abstract: pls tell the information about the genetic characteristics of the two loci, and the Haplotype and its combination on the growth traits.

Response: Thanks for your good suggestion. I have rewritten the abstract.

  1. L33: both loci have significant associations for all parameters? Seems not

Response: I’m sorry for my mistake. I have rewritten the abstract.

  1. L35 compared with

Response: I’m sorry for my mistake. It has revised in the manuscript.

  1. L37 body height

Response: I’m sorry for my mistake. It has revised in the manuscript.

  1. L38 the incomplete conclusion, the sentence is not completed

Response: I’m sorry for my mistake. I have rewritten the abstract.

  1. L63 jak2 change to JAK2

Response: I’m sorry for my mistake. It has revised in the manuscript.

  1. L86 repeated “goat”

Response: I’m sorry for my mistake. I have rewritten the sentence.

  1. L102 pls give a brief introduction of the relationship between the body size and its index from the different references?

Response: Thanks for your good suggestion. It has revised in the manuscript.

Line 107-113. “As recommended by the textbook written by Luo and Wang (1998) [25], the following metrics were also computed: trunk index (TI, Chest circumference/Body length × 100), body length index (BLI, Body length/Body height × 100), chest circumference index (ChCI, Chest circumference/Body height × 100), cannon circumference index (CaCI, Cannon circumference/Body height × 100), chest width index (CWI, Chest width / Chest depth × 100), and huckle bone width index (HuWI, Chest width / hucklebone width × 100).”

  1. L131and L143: pls explain the Statistical analysis of the combined genotypes with the growth traits

Response: Thanks for your good suggestion. The information of statistical analysis between the combined genotypes with the growth traits was added in “2.5. Statistical analyses”.

  1. L191: Table 3. The accession numbers and details of the amino acid sequences employed in constructing the tree, this table is not necessary and can be deleted. The authors can describe the length of the AA sequences in the text.

Response: Thanks for your good suggestion. It has revised in the manuscript.

Line 194-205. “The goat JAK2 mRNA sequence was found to encode a protein consisting of 1132 amino acids. We observed that the coding sequence of goat JAK2 shares 99%, 99%, 99%, 99%, 99%, 96%, 93%, 94%, 96%, and 94% similarity with Ovis aries (XP_042099355.1, 1132 amino acids), Bos taurus (XP_005210038.1, 1132 amino acids), Bos indicus (XP_019821553.1, 1132 amino acids), Bos mutus (ELR59526.1, 1132 amino acids), Bubalus bubalis (XP_006068266.1, 1132 amino acids), Camelus dromedarius (XP_031305623.1, 1130 amino acids), Mus musculus (NP_032439.2, 1132 amino acids), Rattus rattus (XP_032747668.1, 1132 amino acids), Sus scrofa (XP_020938465.1, 1131 amino acids), and Homo sapiens (NP_001309125.1, 1132 amino acids). Phylogenetic analysis conducted using the goat JAK2 sequence revealed a close evolutionary relationship with sheep JAK2, while indicating the least similarity with the clade embodying sequences from R. rattus and M. musculus (Figure 2).”

  1. L195: Individuals with ID genotypes were larger than those with ID with ...? what does the author want to tell the readers? not complete or clear descriptions, which is also not consistent with those in the table 4.

Response: Thanks for your good suggestion. I have rewritten this sentence.

Line 213-216. “For the del19008 locus, individuals with ID genotypes displayed a more superior growth trait than DD with BH (P = 0.014) and TI (P = 0.009) in the group of NB, and with HuW (P = 0.013) in the FQ breed, as well as CWI (P = 0.019) in the JZ population.”

  1. L228: Table 5, regarding to the letters indicating the differences, they are not marked correctly. eg. the first line (FQ, ChW), should be AB, B, Aa, ABb? Pls check them especially for the FQ breed, BH and HuWI in NB breed, and explain it in the legends: the same letters or no letters indicate no significant difference.

Response: I’m sorry for my mistake. It has revised in the Table 4.